optics/nanotechnology/materials science

wafer-scale crystalline carbon nanotubes, controlled vacuum filtration, photonics, optoelectronics

**Author for correspondence:**
Weilu Gao
e-mail: weilu.gao@rice.edu

# Science and applications of wafer-scale crystalline carbon nanotube films prepared through controlled vacuum filtration

## Weilu Gao[1] and Junichiro Kono[1,2,3]

[1]Department of Electrical and Computer Engineering, [2]Department of Physics and Astronomy, and [3]Department of Materials Science and NanoEngineering, Rice University, Houston, TX 77005, USA

WG, 0000-0003-3139-034X; JK, 0000-0002-4195-0577

Carbon nanotubes (CNTs) make an ideal one-dimensional (1D) material platform for the exploration of novel physical phenomena under extremely strong quantum confinement. The 1D character of electrons, phonons and excitons in individual CNTs features extraordinary electronic, thermal and optical properties. Since their discovery in 1991, they have been continuing to attract interest in various disciplines, including chemistry, materials science, physics and engineering. However, the macroscopic manifestation of 1D properties is still limited, despite significant efforts for decades. Recently, a controlled vacuum filtration method has been developed for the preparation of wafer-scale films of crystalline chirality-enriched CNTs, and such films have enabled exciting new fundamental studies and applications. In this review, we will first discuss the controlled vacuum filtration technique, and then summarize recent discoveries in optical spectroscopy studies and optoelectronic device applications using films prepared by this technique.

This article has been edited by the Royal Society of Chemistry, including the commissioning, peer review process and editorial aspects up to the point of acceptance.

## 1. Introduction

Carbon is a fundamental element in Group IV of the periodic table of elements, which is not only essential for life on earth but also promising for modern technologies in communication, computation, imaging, sensing, energy, health and space. Each carbon atom has six electrons, occupying the $1s$, $2s$ and $2p$ atomic orbitals. Two core electrons ($1s^2$) are strongly bound, while four weakly bound electrons ($2s^2 2p^2$) can form covalent bonds in carbon materials using $sp^n$ ($n = 1, 2, 3$) hybrid orbitals. Various

**Figure 1.** The crystal and band structure of SWCNTs. (*a*) Real space and (*b*) reciprocal space representation of the hexagonal lattice of 2D graphene. $a_{C-C}$ is the carbon−carbon bond length. $\mathbf{a}_{1,2}$ and $\mathbf{b}_{1,2}$ are the primitive unit vectors in real and reciprocal space, respectively. The dashed line parallelogram defines the unit cell of graphene with two different carbon atoms A and B. High-symmetry points in reciprocal space, $\Gamma$, M and K, are labelled in the 1st Brillouin zone. (*c*) A chiral vector $\mathbf{C}$ of a SWCNT, defined on the graphene sheet as a linear combination of $\mathbf{a}_1$ and $\mathbf{a}_2$ with integer coefficients. An index (*n,m*) represents SWCNT chirality. Different index combinations result in either (*d*) a metallic SWCNT or (*e*) a semiconducting SWCNT.

hybridization states in carbon lead to a versatile chemical nature and determine the properties of carbon allotropes and organic compounds [1,2]. Two natural carbon allotropes, diamond and graphite with $sp^3$ and $sp^2$ hybridization, respectively, show distinct properties; they were the only known carbon allotropes for a long time until the discovery of zero-dimensional (0D) $C_{60}$ in 1985 [3], giving birth to nanotechnology. After the advent of $C_{60}$, a growing family of carbon nanomaterials emerged, including 1D carbon nanotubes (CNTs) [4] and two-dimensional (2D) graphene [5]. Still other members, based on numerous combinations and modifications of carbon atom hybridization, are yet to be discovered. Despite the unique properties of each member of the carbon nanomaterial family, their atomic structures are related to one another. In this sense, graphene can be viewed as a fundamental building block because fullerenes, carbon nanotubes and graphite can all be derived from graphene [6]. In particular, a single-wall CNT (SWCNT) is a rolled-up version of monolayer graphene, which possesses unique 1D thermal, mechanical, electronic and optical properties [1,7–9].

## 1.1. Electronic, optical properties and applications of single-wall carbon nanotubes

The crystal and band structures of SWCNTs are highly related to those of graphene. However, the strong confinement of electronic states along the tube circumference makes SWCNT a completely different material. Graphene consists of a triangular lattice of $sp^2$-bonded carbon atoms, as shown in figure 1*a*. The carbon−carbon bond length $a_{C-C}$ is 1.42 Å. The primitive unit cell contains two carbon atoms, A and B, contributing two $\pi$ electrons that play fundamental roles in electrical and optical phenomena. Figure 1*b* shows the reciprocal lattice of graphene, which is also a triangular lattice, and there are three important high-symmetry points: $\Gamma$-point (the centre of the Brillouin zone), *M*-point and *K*-point. The $\mathbf{a}_{1,2}$ and $\mathbf{b}_{1,2}$ are the primitive vectors in real and reciprocal space, respectively.

The band structure of SWCNTs sensitively depends on the manner in which the graphene sheet is rolled up, which can be uniquely specified by defining the chiral or roll-up vector $\mathbf{C} = n\mathbf{a}_1 + m\mathbf{a}_2$,

where $n$ and $m$ are positive integers with $n \geq m$ (figure 1*c*). A SWCNT is formed by wrapping the graphene sheet in such a way that the two points connected by **C** meet, with a diameter $d_t$. The chiral indices $(n,m)$ fully determine the crystal structure of the SWCNT, and the character of the band structure depends on the value $v = (n - m) \bmod 3$:

(1) If $v = 0$, the SWCNT is metallic;
(2) If $v = 1$ or 2, the SWCNT is semiconducting with a band gap $\sim 0.7$ eV$/d_t$ (nm).

However, when the nanotube's curvature is taken into account, only armchair nanotubes ($n = m$) are truly metallic. For other metallic tubes with $v = 0$ and $n \neq m$, there is a small, curvature-induced band gap that scales as $1/d_t^{\,2}$.

As shown in figure 1*d*,*e*, SWCNTs exhibit a series of peaks in density of states at subband edges, known as van Hove singularities, characteristic of 1D systems; this significantly concentrated density of states leads to strongly enhanced optical absorption and emission processes in the vicinity of subband edges. In addition, the strong quantum confinement of electrons and holes results in the generation of stable 1D excitons, hydrogen-like bound electron-hole pairs; these excitons can be excited either optically or electrically, and their radiative recombination results in photoluminescence or electroluminescence, respectively. Furthermore, the 1D nature of SWCNTs generally restricts carrier scattering to occur only along the nanotube axis while in a higher dimensional material system scattering directions can be arbitrary. In particular, the unique band structure of metallic SWCNTs near the Dirac point leads to a suppression of back scattering and a very long mean free path [10], which are ideal for quantum wire interconnections [11]. Furthermore, field effect transistors built from semiconducting SWCNTs have demonstrated large on/off ratios and extremely high room temperature mobilities (greater than $10^5$ cm$^2$ Vs$^{-1}$ [12]). In addition, the optical absorption of semiconducting SWCNTs can be electrically modulated, laying the foundation for optoelectronic device applications.

Although individual CNTs provide an ideal platform for investigating fundamental properties of CNTs and demonstrating proof-of-concept optoelectronic device applications, they are not suited for practical applications for various reasons, including complicated fabrication processes required, low photoluminescence quantum yields, and most importantly, the absence of methods of scaling up [13]. Individual CNT transistors are considered to be impractical due to the difficulty in positioning CNTs precisely, which applies to individual CNT photonic and optoelectronic devices as well. Light-matter interaction in individual CNTs is unusually strong, and further cooperative enhancement can be expected when a large number of CNTs can be assembled in an ordered manner.

Successfully assembling individual CNTs on a macroscopic scale, in the form of random networks or aligned arrays, is the key to facilitating practical applications. An ensemble assembly can help overcome challenges in individual CNT devices by (i) minimizing device variations, (ii) providing a large active area, and (iii) not requiring precise individual tube positioning. Currently, there are various ways to prepare a CNT ensemble, including chemical vapour deposition (CVD) [14,15] and solution-based deposition [16]. The produced ensemble can exhibit either a semiconducting or metallic material behaviour and can be useful for various applications. For example, CNTs are thermally and chemically stable [17], making them ideal for high-temperature emitters [18] (figure 2*a*) and compatible with high-$\kappa$ material deposition for high-performance electronics [19] (figure 2*b*). A large ratio of the surface area to the volume of CNTs enables an efficient charge transfer doping from surrounding chemicals for optimized thermoelectric materials [20] (figure 2*c*). The excellent mechanical strength of CNTs is perfect for large-area flexible electronic/optoelectronic applications [21] (figure 2*d*). Large active areas possess the potential for an efficient broadband detector [22,25] (figure 2*e*) and enable low-cost, high-efficiency heterogeneous solar cells [23] (figure 2*f*). Furthermore, a highly conductive metallic SWCNT thin film of a small optical absorption cross-section with strong mechanical properties can serve as a stretchable transparent electrode, which is important for certain security applications and backlit displays [24] (figure 2*g*). CNTs can also form composites with other materials, such as organic polymers and inorganic nanorods, to reinforce polymer mechanical properties for nano/micro-electro-mechanical systems [26] and build sensitive gas sensors [27].

To best preserve the extraordinary properties of individual CNTs, macroscopic ensembles with a single domain of perfectly aligned CNTs, i.e. crystalline CNTs, are highly preferred. In SWCNT transistors, for example, the composition and morphology of conducting channels strongly affect the device performance. The degree of SWCNT alignment and the purification of semiconducting SWCNTs affect two important device parameters: the carrier mobility and the on/off ratio [28]. With a perfectly aligned semiconducting SWCNT ensemble, the mobility and on/off ratio of a large-scale

**Figure 2.** Representative photonic and optoelectronic devices using CNTs. (*a*) A current-driven high-speed blackbody light emitter [18]. (*b*) A high-performance transistor [19]. (*c*) Thermoelectric materials based on semiconducting SWCNTs with tailored doping profiles [20]. (*d*) Medium-scale flexible thin-film transistors [21]. (*e*) A broadband photothermoelectric detector [22]. (*f*) High-efficiency solar cells based on SWCNT-silicon heterojunctions [23]. (*g*) Large-scale transparent electrodes [24].

transistor are expected to be comparable with transistor devices based on individual CNTs. By contrast, a randomly oriented ensemble generally has a lower mobility due to stronger intertube scattering [13]. Furthermore, even a small amount of metallic nanotubes (e.g. 1%) dramatically decreases the on/off ratio (e.g. three orders) because of the percolation path formed by metallic tubes [29]. Therefore, it is crucial to separate nanotubes by electronic types and assemble them in an ordered manner to extend the excellent properties of individual CNTs to CNT macroscopic ensembles for real-world applications.

## 1.2. Challenges in synthesis of wafer-scale crystalline carbon nanotubes

Today, it is still challenging to produce a macroscopic single-domain, highly aligned, densely packed and chirality-enriched SWCNT film. Current assembly techniques can be separated into two general categories: (i) *in situ* direct-growth assembly methods and (ii) *ex situ* post-growth assembly methods. However, techniques in both categories have limitations.

Direct-growth assembly methods mainly consist of the CVD growth method for either vertical [14,30–32] or horizontal alignment [33] with respect to the growth substrate. If the catalyst density is high enough, a crowding effect leads to vertical alignment of CNTs with heights as large as mm, as shown in figure 3*a*. The as-grown patterns of well-aligned SWCNTs synthesized using the CVD method can be laid down to form horizontally aligned patterns and be transferred to any substrates through a scalable, facile and dry approach [36]. To directly assemble horizontally aligned arrays of CNTs, multiple external-stimuli-assisted CVD growth methods have been developed, including electric field [34], gas flow [35] and epitaxy [19]. Electric fields can orient CNTs because of the large and highly anisotropic polarizability of CNTs (figure

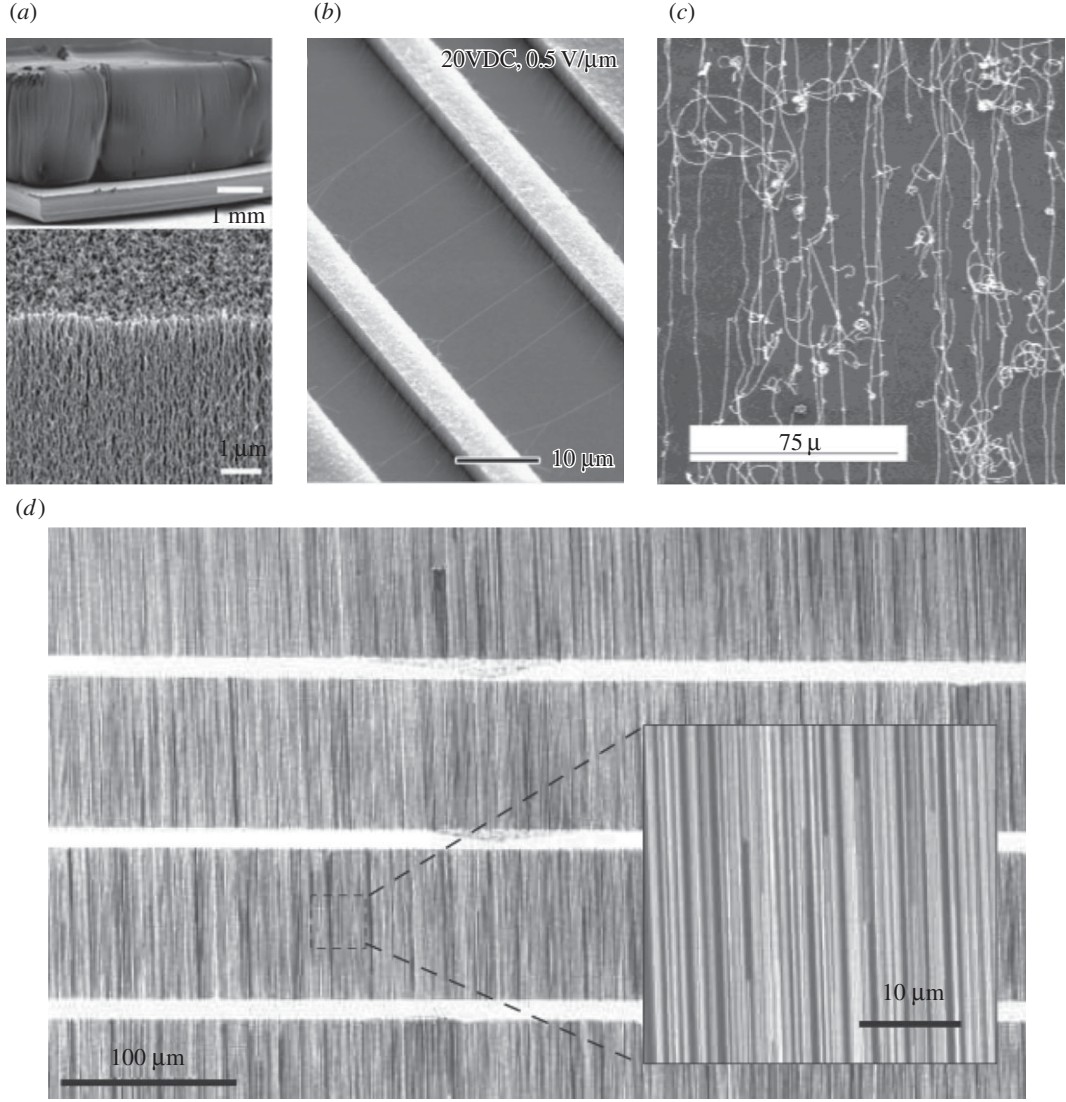

**Figure 3.** *In situ* direct-growth assembly methods of CNTs. (*a*) Water-assisted CVD growth [31]. (*b*) Electric field directed CVD growth [34]. (*c*) Gas flow directed CVD growth [35]. (*d*) Epitaxy CVD growth [19].

3*b*); a feeding gas flow can generate a flow field that can carry CNTs to overcome the gravity to grow along the flow direction (figure 3*c*); and the oriented van der Waals force along the substrate surface can produce horizontally aligned CNTs (figure 3*d*).

However, there is a common issue for direct-growth methods: it is still very challenging to control the chirality during the growth procedure. Obtained aligned arrays have mixed electronic types and are not desirable for most applications. Recently, there have been breakthroughs in precisely controlling chirality distribution during synthesis, by designing a catalyst that matches the carbon atom arrangement around the nanotube circumference of a specific chirality [37–39] or applying an external electric field to control the polarity of catalyst charge for changing the chirality from metallic to semiconducting [40]. However, the packing density of aligned arrays is relatively low. The highest density reported is approximately $50$ CNTs $\mu m^{-1}$ [41], which is quite dilute in a practical sense, especially for photonic and optoelectronic applications.

Post-growth assembly methods mainly consist of solution-based techniques for CNT alignment [42]. These methods can take advantage of solution-based SWCNT separation techniques, such as the DNA-wrapping method [43,44], the polymer-wrapping method [45], density gradient ultracentrifugation [46,47], gel chromatography [48] and aqueous two phase extraction [49,50]. Existing post-growth assembly techniques use external force, self-assembly and liquid crystal phase transition.

External forces used in CNT alignment can be mechanical, magnetic and electric. The simplest method is based on mechanical force by dry-spinning CNTs from a CVD-grown CNT forest (figure 4*a*) [51]; however,

royalsocietypublishing.org/journal/rsos　R. Soc. open sci. **6**: 181605

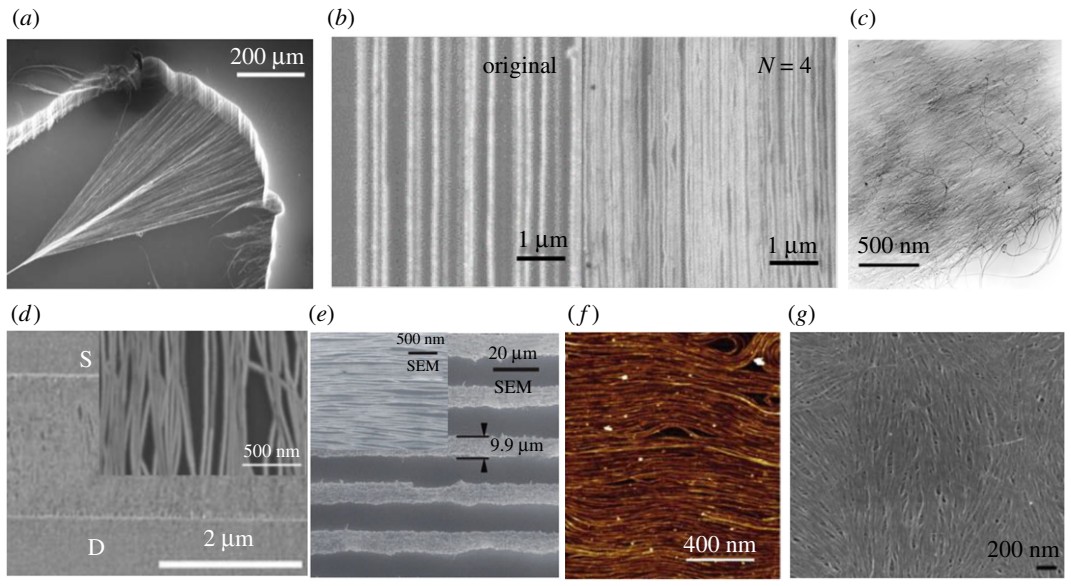

**Figure 4.** *Ex situ* post-growth assembly methods of CNTs. (*a*) Dry-spinning after water-assisted CVD growth [51]. (*b*) Direct shrinkage transfer for four times ($N = 4$) density amplification of aligned CNTs [52]. (*c*) Magnetic field induced alignment [53]. (*d*) Dielectrophoresis [54]. (*e*) Evaporation-driven self-assembly [55]. (*f*) The Langmuir–Blodgett method [56] and (*g*) Liquid crystal phase transition of CNTs [57].

there is no chirality selectivity in this method. Metallic CNTs in horizontally aligned CNTs prepared using epitaxy CVD growth can be selectively removed, leaving an array of high-purity semiconducting CNTs [58]. However, the packing density is as low as 3 CNTs $\mu m^{-1}$. A simple directional shrinkage method can compress films to amplify the packing density (figure 4*b*) [52]. Although this level of low packing density is desirable for high-performance electronic devices applications, such densities are too low for photonic and optoelectronic applications. The same technique can be applied to randomly oriented CNT films to reduce the angle deviation for preparing partially aligned films [59]. An applied magnetic field can align CNTs based on their magnetic susceptibility anisotropy, but the required magnetic field to achieve meaningful alignment (figure 4*c*) is typically larger than 10 T [53], unsuitable for real-world applications. A more affordable and practical method is to align CNTs in liquid by applying an AC electric field, called dielectrophoresis [54], as shown in figure 4*d*. The obtained films have alignment packing densities up to 30–50 CNTs $\mu m^{-1}$, which is quite low. Furthermore, the alignment quality is relatively poor, and full surface coverage is nearly impossible.

Instead, self-assembly methods, including evaporation-driven self-assembly [55,60], Langmuir-Blodgett [56] and Langmuir-Shaefer [61] assembly, can produce a large-sized full-surface-coverage aligned film. As the substrate is immersed inside a CNT suspension during the evaporation-driven self-assembly process, the interplay between the surface friction force, liquid surface tension and capillary force can align CNTs at the solid–liquid–air interface (figure 4*e*). Similarly, when the immersed substrate is pulled out of the CNT suspension vertically (Langmuir-Blodgett) or horizontally (Langmuir-Shaefer), the 2D confinement at the air–water interface causes CNTs to orient along certain directions (figure 4*f*). A semiconducting SWCNT aligned monolayer film of an impressively high packing density of approximately 500 CNTs $\mu m^{-1}$ with full surface coverage has been demonstrated for significantly improved transistor performance [61]. However, the thickness of fabricated films using these techniques is limited to a few nanometres because of the 2D nature of these techniques, and again the degree of alignment is not very high.

By contrast, a liquid crystal phase transition can lead to alignment in three dimensions. One strategy is to mix CNTs with a liquid crystal polymer matrix [62,63]. Under certain conditions, the liquid crystal polymer can go through a phase transition spontaneously with macroscopically ordered structures, where the hosted CNTs follow the same transition. The degree of alignment can be very high, but the challenge is how to remove the liquid crystal polymers completely without destroying the alignment of CNTs. As a consequence, it motivated another strategy to use the liquid crystal transition of CNTs based on Onsager's theory. Different aqueous dispersion systems have been investigated to achieve nematic ordering of CNTs [57,64,65] in a short range (figure 4*g*), but wafer-scale crystalline CNT films through a liquid crystal phase transition have not been achieved.

# 2. Fabrication and characterization of wafer-scale crystalline films of chirality-enriched carbon nanotubes

Post-growth assembly methods are suitable for fabricating 3D architectures of macroscopically aligned CNTs, because of the availability of various solution-based chirality separation techniques. However, the capabilities of existing methods are limited in packing density, thickness, and the spatial scale of domains in which nanotubes are aligned. Currently, there are no available techniques for fabricating large-area monodomain films of highly aligned, densely packed and chirality-enriched SWCNTs with controllable thickness. Recently, He *et al.* have developed a new solution-based alignment technique based on controlled vacuum filtration [66], which can provide a uniform, wafer-scale SWCNT film of an arbitrarily and precisely controllable thickness (from a few nm to approximately 100 nm) with a high degree of alignment (nematic order parameter $S \sim 1$) and packing (approx. $3.8 \times 10^5$ tubes in a cross-sectional area of 1 $\mu m^2$) in a simple, well-controlled and reproducible manner, regardless of the synthesis method, type or chirality of the SWCNTs used. Furthermore, the produced films are compatible with standard micro/nanofabrication processes used to fabricate various electronic and photonic devices.

The controlled vacuum filtration process starts with a well-dispersed CNT suspension, where individual CNTs are separated and suspended. To prepare an aqueous suspension, the sidewalls of SWCNTs have to be functionalized by either chemical functional groups or amphiphiles with a hydrophobic tail and a hydrophilic head [67]. Ultrasonication is a widely used method for general nanoparticle dispersion, agitating molecules in either a bath or probe (or horn) sonicator. The generated ultrasound wave is delivered to a CNT bundle in a surfactant solution, and individual CNTs are peeled off from the outer part of the CNT bundle and wrapped by surfactant molecules to prevent future rebundling. The obtained suspension is then ultracentrifuged to remove any undissolved bundles, and the supernatant is collected. The prepared suspension is poured into a vacuum filtration system, as shown in figure 5*a*. A filter membrane with a pore size smaller than the CNT length blocks the passage of CNTs and allows water to penetrate through, under a differential pressure applied across the membrane that is maintained by a vacuum pump underneath. During the deposition process of CNTs, the penetrability of the filter membrane reduces gradually. This self-limiting process guarantees a uniform film [24]; (figure 5*b*). The obtained film can be readily transferred onto any substrate, such as a quartz wafer in figure 5*c*, by dissolving the filter membrane in organic solvents. Furthermore, the controlled vacuum filtration method can be applied to any CNT species, as long as the CNTs are well dispersed, enabling us to take advantage of solution-based chirality separation techniques. For example, filtrated films made from mono-dispersed SWCNTs separated using density gradient ultracentrifugation show different colours due to subtractive coloration [70].

Normally the vacuum filtration technique can only produce randomly oriented CNTs. Figure 5*e* shows an atomic force microscopy (AFM) image of randomly oriented films of CNTs prepared in such a standard way. Previous work [57] revealed some local ordering utilizing the liquid crystal phase transition of CNTs, as CNTs accumulate near the surface of the filter membrane during the filtration process. They empirically identified three crucial conditions to achieve macroscopically aligned films by vacuum filtration: (i) the surfactant concentration must be below the critical micelle concentration; (ii) the CNT concentration must be below a threshold value; and (iii) the filtration process must be well controlled at a low speed. Furthermore, the surface chemistry of the filter membrane and CNT structural parameters have an important influence on optimal conditions for macroscopic alignment. The scanning electron microscopy (SEM) image in figure 5*f* shows aligned SWCNTs, and the cross-sectional transmission electron microscopy (TEM) image in figure 5*g* shows ordered SWCNT structure. Furthermore, Falk *et al.* [69,71] reproduced the work of He *et al.* [66] and performed studies of crystal structures in obtained films using cross-sectional TEM and X-ray diffraction analyses. Cross-sectional TEM images in figure 5*h,i* show that densely packed CNTs assembled in controlled vacuum filtration form a crystalline structure, and selected-area TEM diffraction patterns in figure 5*j* confirm a triangular lattice. Figure 5*k* displays the X-ray diffraction spectra for both aligned and randomly oriented films, and a prominent peak at $2\theta = 5.8°$ is only present in the aligned film. This peak corresponds to a lattice constant of 1.74 nm, an excellent agreement with the summation of the average diameter of CNTs (1.4 nm) and the constant spacing between carbon atoms (0.34 nm). Based on this information, the packing density in these films was estimated to be approximately $3.8 \times 10^5$ tubes in a cross-sectional area of 1 $\mu m^2$ [69].

Quantitative characterization of alignment quality can be made by calculating the nematic order parameter, *S*. It is defined as the ensemble average value of $(3\cos^2\theta - 1)/2$ in 3D and $2\cos^2\theta - 1$ in 2D,

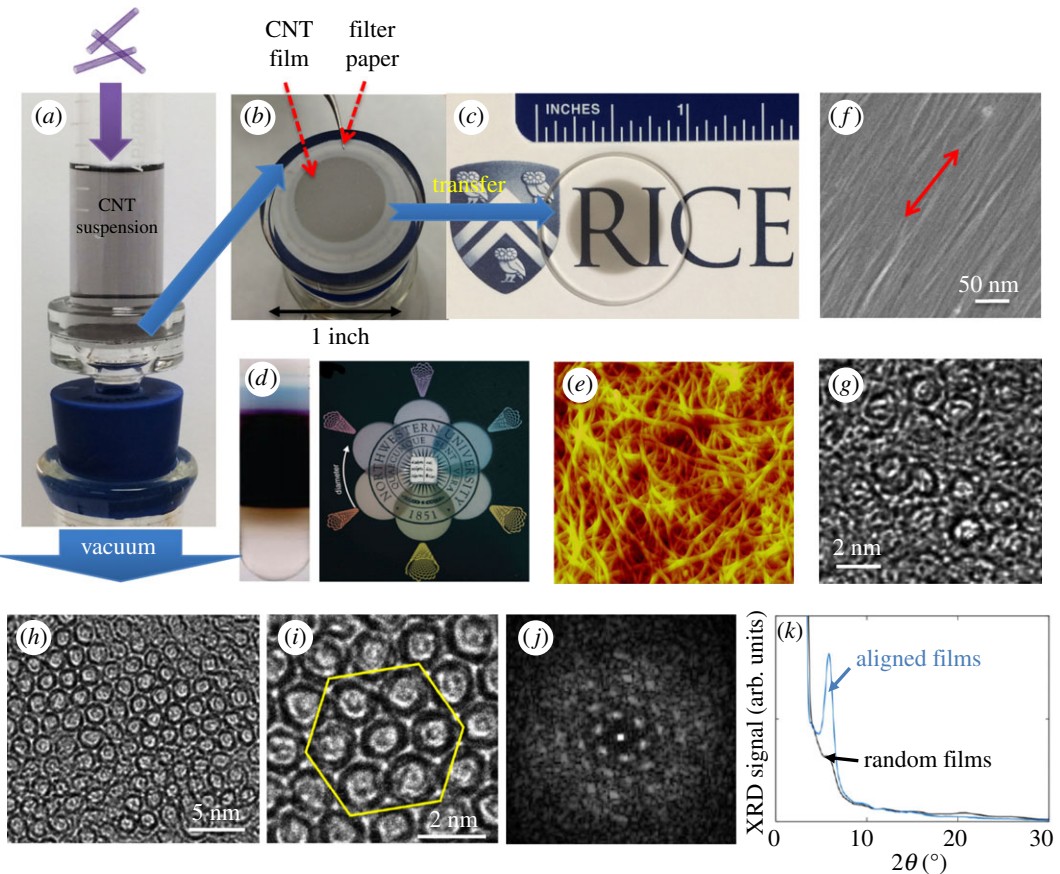

**Figure 5.** Wafer-scale crystalline CNT films fabricated using controlled vacuum filtration. (*a*) A CNT suspension goes through a vacuum filtration system. (*b*) A wafer-scale, uniform CNT film is formed on the filter membrane. (*c*) Optical image of the produced film after being transferred onto a transparent substrate. (*d*) Separated mono-dispersed suspension using the density gradient ultracentrifugation method and fabricated semitransparent colourful films [68]. (*e*) A typical AFM image of random networks of CNTs fabricated using the conventional vacuum filtration technique [24]. (*f*) A high-resolution SEM image and (*g*) a cross-sectional TEM image of the film. Adapted from [66]. (*h,i*) Cross-sectional TEM images of a film produced by Falk and coworkers using the controlled vacuum filtration technique, showing crystalline CNTs with a high packing density approximately $3.8 \times 10^5$ CNTs $\mu m^{-2}$. (*j*) A TEM diffraction image of a selected area in (*i*), showing a hexagonal lattice. (*k*) Grazing-incidence X-ray diffraction spectrum of a crystallized CNT film (blue curve), and a control film of randomly oriented CNTs (black curve). Adapted from [69].

where $\theta$ is the angle deviation from the macroscopic alignment direction. Figure 6*a* shows the angle distribution of SWCNTs in an obtained film made of arc-discharge SWCNTs with an angle standard deviation of approximately 1.5° across an area of approximately 1 mm², indicating that $S$ is approximately 1. This densely packed and aligned SWCNT film exhibits strongly anisotropic optical attenuation in the entire electromagnetic spectrum, from the terahertz (THz) to the visible, as shown in figure 6*b*. Specifically, there is no attenuation for the perpendicular polarization in the entire THz/ infrared range (less than 1 eV), whereas there is a prominent, broad peak at approximately 0.02 eV in the parallel direction because of the plasmon resonance in finite-length CNTs [72]. Figure 6*c* plots the same spectra in the infrared range with the energy axis on a linear scale. The two interband transitions in semiconducting nanotubes ($E_{11}^S$ and $E_{22}^S$) and the first interband transition in metallic nanotubes ($E_{11}^M$) are clearly observed. According to the selection rules that will be discussed later in detail, these peaks are completely absent for the perpendicular polarization case. Instead, a broad absorption feature is observed between the $E_{11}^S$ and $E_{22}^S$ peaks, which can be attributed to the crosspolarized and depolarization-suppressed $E_{12}^S/E_{21}^S$ absorption peak [73–75]. However, due to the mixed electronic types inside this film, the peaks are inhomogeneously broadened, and an investigation of highly aligned single-chirality SWCNTs is necessary for obtaining quantitative information on crosspolarized excitons (see Section 3).

The $S$ parameter can be determined using polarization-dependent THz spectroscopy, through the equations $S = (A_{\parallel} - A_{\perp})/(A_{\parallel} + 2A_{\perp})$ in 3D and $S = (A_{\parallel} - A_{\perp})/(A_{\parallel} + A_{\perp})$ in 2D, where $A_{\parallel}$ is the attenuation along the alignment direction and $A_{\perp}$ is the attenuation perpendicular to the alignment direction. These equations assume that the transition dipole is entirely parallel to the nanotube axis

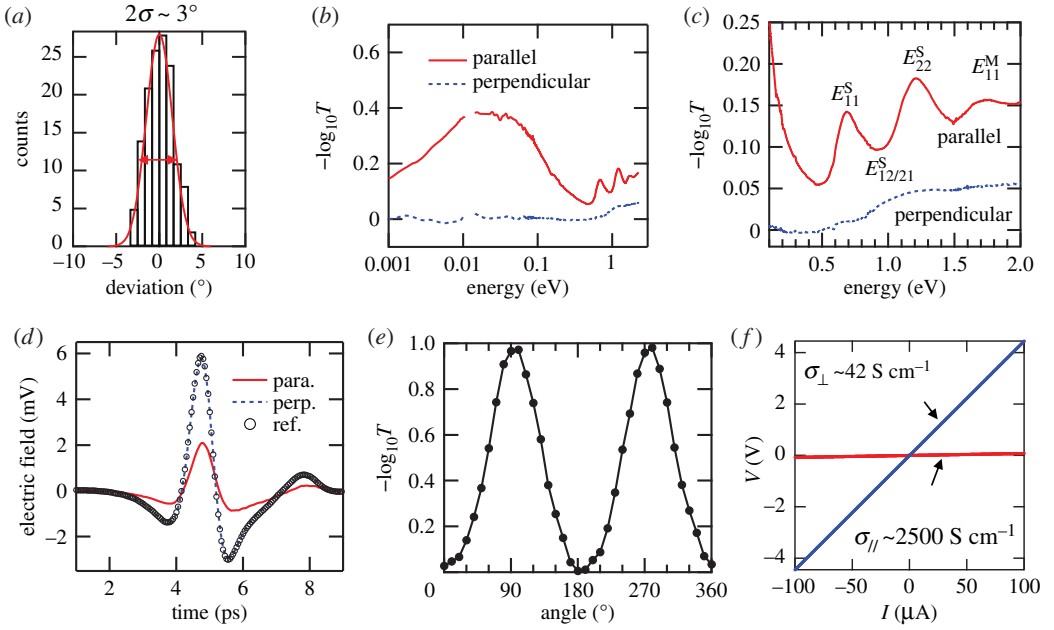

**Figure 6.** Characterization of crystalline films of arc-discharge SWCNTs of mixed electronic types. (a) Angular distribution of CNTs within a 1 cm$^2$ area of the film determined by SEM image analysis. (b) Polarization-dependent attenuation spectra in a wide spectral range, from the THz/far-infrared to the visible. (c) Expanded view of (b), showing interband transitions. (d) Time-domain THz waveforms of transmitted THz radiation for parallel and perpendicular polarizations. (e) Attenuation as a function of the angle between the THz polarization and the nanotube alignment direction. (f) Voltage–current relationship when the current flow is parallel and perpendicular to the CNT alignment direction. Adapted from [66].

direction, which is valid for THz response. Figure 6d shows time-domain waveforms of THz radiation transmitted through an aligned arc-discharge SWCNT film on an intrinsic silicon substrate for polarizations parallel and perpendicular to the alignment direction, together with a reference waveform obtained for the substrate alone. The data for the perpendicular case completely coincide with the reference trace; that is, no attenuation occurs within the SWCNT film. However, there is significant attenuation for the parallel case. Figure 6e shows a more detailed polarization-angle dependence of THz attenuation. The calculated S is approximately 1, which is consistent with SEM analysis. Moreover, the obtained film shows strong anisotropic electronic conductivity (figure 6f), with the conductivity along the alignment direction $\sigma_\parallel \sim 2500\,\mathrm{S\,cm}^{-1}$, the conductivity perpendicular to the alignment direction $\sigma_\perp \sim 42\,\mathrm{S\,cm}^{-1}$ and the ratio between these two $\sigma_\parallel/\sigma_\perp \sim 60$ at room temperature.

One of the most important features of controlled vacuum filtration is its universal applicability to any SWCNT suspensions, which allows us to capitalize on solution-based chirality separation techniques. Figure 7a shows two images of separated semiconducting (10,3) and (6,5) mono-dispersed SWCNTs on a large scale, using gel chromatography and aqueous two-phase extraction methods, respectively. Similarly, the fabricated chirality-enriched films show highly aligned and densely packed structures, as demonstrated in figure 7b. As semiconducting SWCNTs are optically pumped on resonance with the second interband transition (figure 1e), there is resonant emission at the energy of the first interband transition. Moreover, the SWCNTs are all aligned along the same direction, and thus the emitted light is linearly polarized, as shown in figure 7c,d.

The performance of transistors made from semiconductor-enriched aligned films (figure 7e) is better than that of their counterparts made from random films. As shown in figure 7f,g, the on-current density of the transistor in the parallel (perpendicular) direction is approximately $2\,\mathrm{nA\,\mu m}^{-1}$ (approx. $80\,\mathrm{pA\,\mu m}^{-1}$), indicating that the on-current density can be improved by aligning the CNTs in one direction. The on-current density can be further enhanced by using larger diameter nanotubes, which is also demonstrated by our transistors based on semiconductor-enriched arc-discharge SWCNTs with an average diameter of 1.4 nm (figure 7g). The device shows an enhancement of on-current density by approximately 50 times compared to the (6,5) CNT transistor at the same drain-source voltage. All fabricated devices display decent on/off ratios approximately $1 \times 10^3$, which can be improved by using a high-$\kappa$ dielectric material or using a top-gate structure.

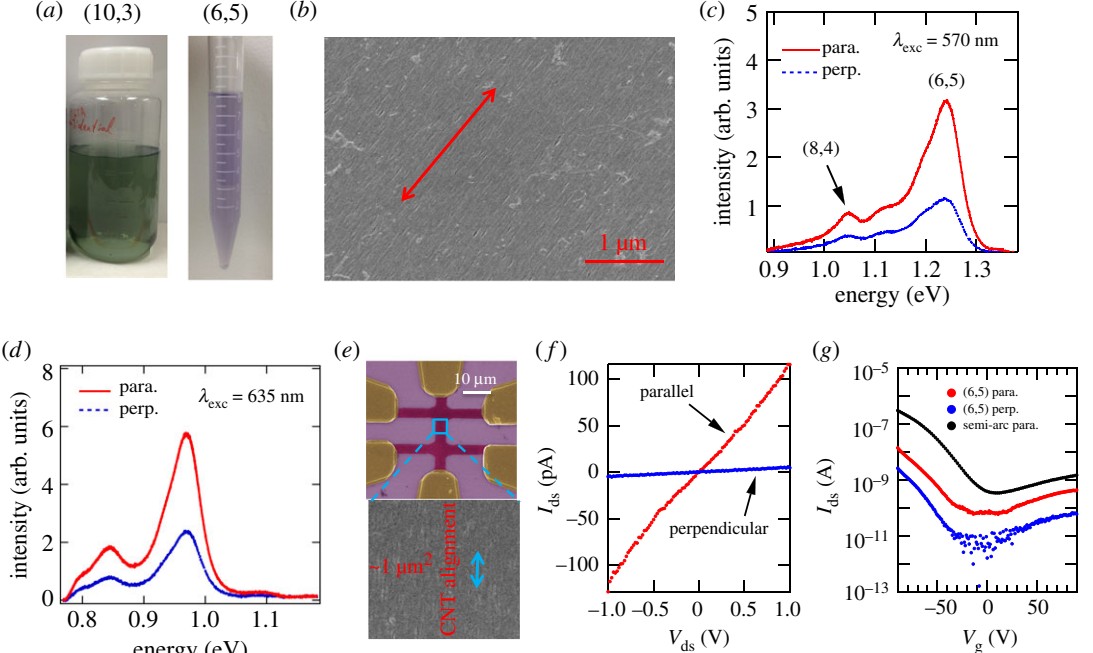

**Figure 7.** Macroscopically aligned chirality-enriched SWCNT films and devices. (*a*) Chirality separated semiconducting (10,3) and (6,5) SWCNTs using gel chromatography and aqueous two-phase extraction methods, respectively. (*b*) A typical SEM image of fabricated aligned chirality-enriched films. Polarized photoluminescence of aligned (*c*) (6,5) and (*d*) (10,3) films, when optically pumped on resonance with the first interband transition. (*e*) False-colour SEM images of a thin-film transistor with a channel width of approximately 5 μm and channel length of approximately 30 μm made from an aligned and (6,5)-enriched SWCNT film. (*f*) Source-drain current versus source-drain voltage at zero gate voltage of the transistor, showing anisotropic conductivities of the aligned (6,5)-enriched thin-film transistor. (*g*) Source-drain current at a source-drain voltage of 1 V versus gate voltage of the (6,5)-enriched transistor. The on-current density is enhanced by a factor of 50 in a transistor made from larger-diameter semiconductor-enriched arc-discharge SWCNTs. Adapted from [66].

Obtained films can be scaled up both vertically and laterally and intercalated by external molecules after multiple-layer stacking to form 3D architectures, as shown by Komatsu *et al.* [76]. By adjusting the filtrated volume during the controlled vacuum filtration process, the thicknesses of obtained films can be simply controlled, and they all display good alignment (figure 8*a*). The extinction ratio (ER) of THz polarized attenuation can be as high as approximately 120 dB μm$^{-1}$, due to the high packing density in the films. However, the lateral size of obtained films is determined by the filtration system size. For instance, a 2-inch system can produce a larger film, compared to 1-inch systems, as shown in figure 8*b*. These 2-inch macroscopic crystalline SWCNT films enable us to build 3D architectures, through both layer stacking and molecule intercalation, as shown in figure 8*c*. This process consists of a simple manual layer stacking with careful control of the angles between layers and chemical doping by either electron donors or acceptors for *n*-type or *p*-type doping, respectively [76]. These films are compatible with a variety of dopants, such as HNO$_3$ [77], H$_2$SO$_4$ [77,78], NH$_4$S$_2$O$_8$ [78], HCl [77], H$_2$SO$_3$ [79], iodine solution [79] and benzyl viologen [80]. Figure 8*d* shows a height profile measured using AFM for four stacked layers, demonstrating this method as a consistent and well-controlled technique. The ER in the THz range increases as the film thickness and doping level increase, as shown in figure 8*e,f*. As the doping level (and thus the Fermi level) is increased, the first interband transition is suppressed and eventually completely quenched (figure 8*g*).

# 3. Absorption of perpendicularly polarized light

Extremely confined 1D excitons with huge binding energies in semiconducting SWCNTs lead to a variety of optical properties [81–85] with light polarized along the SWCNT axis, while there are limited investigations of optical properties for light polarized perpendicular to the SWCNT axis, especially in a macroscopic SWCNT ensemble. The macroscopic crystalline films fabricated through controlled vacuum filtration provide a unique opportunity for such studies. In this section, two examples—intersubband plasmons and crosspolarized excitons—are described.

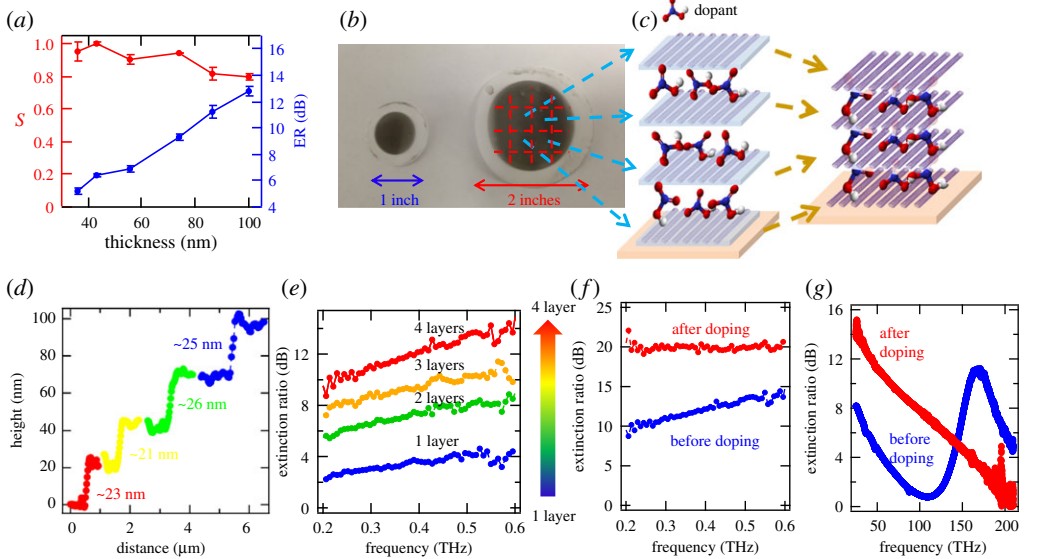

**Figure 8.** 3D architectures made of aligned SWCNT films. (*a*) Nematic order parameter (*S*, left axis) and the extinction ratio (ER, right axis), as a function of film thickness. (*b*) 1-inch and 2-inch aligned films made using controlled vacuum filtration. (*c*) A multilayer structure by stacking thin films of aligned SWCNTs and sandwiching doping molecules in-between. (*d*) Height profile of stacked films from one layer to four layers. (*e*) ER versus frequency in the THz range for 1–4 layers. (*f*) ER versus frequency in the THz range before and after doping. (*g*) ER versus frequency in the infrared range before and after doping. Adapted from [66,76].

Optical selection rules govern the observable transitions under far-field excitation. For light polarized parallel and perpendicular to the SWCNT axis, selection rules are different. Figure 9*a* schematically shows the band structure of metallic and semiconducting SWCNTs, respectively. Arrows of different colours show representative allowed optical transitions for light polarized along the tube axis (blue) and perpendicular to the tube axis (red and yellow). Conduction and valence subbands in metallic (semiconducting) nanotubes are indicated as $C_{Mi}$ and $V_{Mi}$ ($C_{Si}$ and $V_{Si}$), respectively, where $i = 1, 2, 3, \ldots$. Each subband has a well-defined angular momentum with quantum number $n$ and an eigenvalue of the Pauli matrix $\sigma_x$ [74,89]. Light with polarization parallel to the nanotube axis can cause transitions with $\Delta n = 0$. For light polarized perpendicular to the nanotube axis, the following selection rules have to be satisfied:

(1) $\Delta n = \pm 1$, meaning that the angular momentum quantum number has to change by $\pm 1$.
(2) $\Delta\sigma_x = 0$, meaning that the eigenvalue of the Pauli matrix $\sigma_x$ has to be conserved.

In undoped or lightly doped SWCNTs, optical absorption spectra are dominated by *i*-conserving transitions (for example, the $M_{11}$, $S_{11}$ and $S_{22}$ transitions) for parallel polarization. Transitions for perpendicular polarization, such as the lowest energy transitions $M_{12}$ and $S_{12}/S_{21}$, are strongly suppressed due to the depolarization effect [73,75]. These transitions generate crosspolarized excitons. In heavily doped SWCNTs, where the Fermi level ($E_F$) is inside the conduction (valence) band, perpendicularly polarized light can excite the $C_{M1} \rightarrow C_{M2}$ ($V_{M1} \rightarrow V_{M2}$) transition in metallic SWCNTs and the $C_{S1} \rightarrow C_{S3}$ and $C_{S2} \rightarrow C_{S4}$ ($V_{S1} \rightarrow V_{S3}$ and $V_{S2} \rightarrow V_{S4}$) transitions in semiconducting SWCNTs. As has been shown in III-V semiconductor quantum wells, such as GaAs quantum wells [90], these intersubband transitions are intrinsically collective, deviating from the single-particle picture and involving many-body effects, and are thus termed intersubband plasmons (ISBPs). Unlike ISBPs in quantum wells with oblique incident light excitation, ISBPs in 1D SWCNTs can be excited under normal incidence. Also, different from crosspolarized excitons, ISBPs are expected to be strong due to the concentrated joint density-of-states (figure 9*b*) as well as the suppression of the depolarization effect at high $E_F$ [91,92].

Previous studies of heavily doped SWCNTs have revealed a new peak [86,93–95] in absorption spectra. For example, as SWCNT films are heavily doped by $AuCl_3$, a new peak emerges at approximately 1.12 eV, as shown in figure 9*c*. However, in these cases, the nanotubes were randomly oriented, precluding definitive interpretation of the origin of this peak. On the other hand, crosspolarized excitons are significantly suppressed by the depolarization effect, and it is challenging to observe such features in single-CNT spectroscopy and studies of randomly oriented CNTs. For example, crosspolarized photoluminescence of a CNT suspension [87] (figure 9*d*) and circular

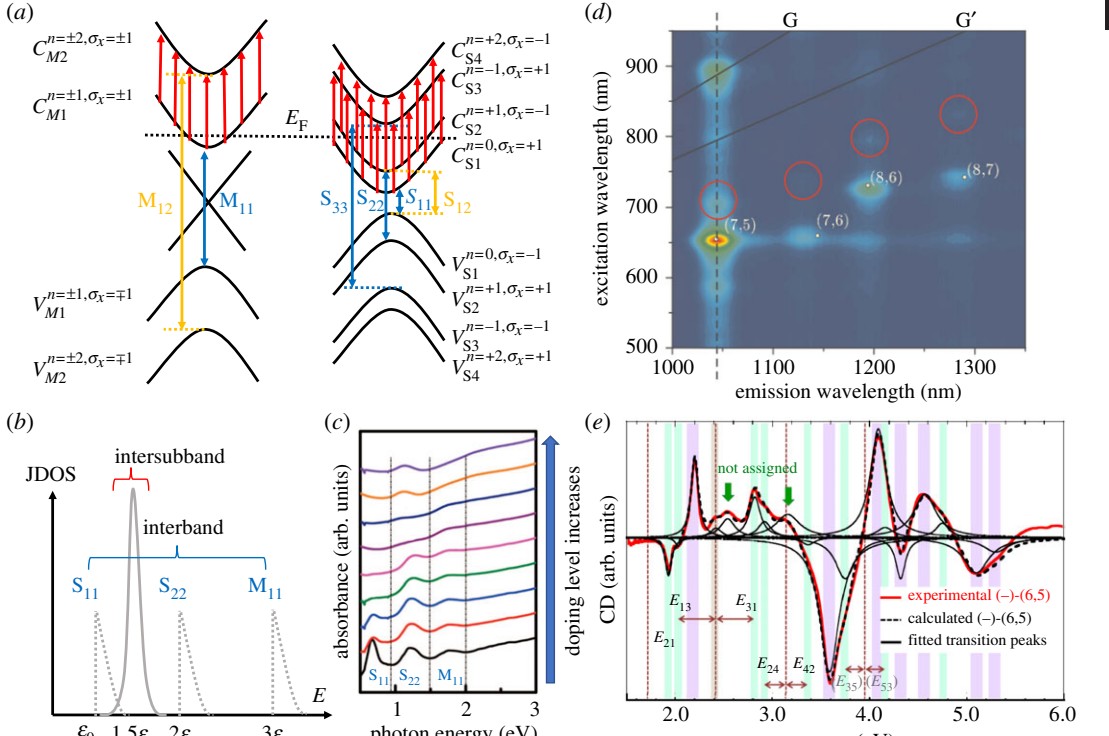

**Figure 9.** Optical properties for light polarized perpendicular to the CNT axis. (*a*) Interband transitions for parallel polarization (blue arrows), interband transitions (yellow arrows) and intersubband transitions (red arrows) for perpendicular polarization in metallic (left) and semiconducting (right) SWCNTs. (*b*) The joint density of states for interband and intersubband transitions. (*c*) UV – vis – near infrared optical absorption spectra of an arc-discharge SWCNT film doped with increasing $AuCl_3$ concentration [86]. (*d*) Cross-polarized photoluminescence of a CNT suspension [87]. (*e*) Circular dichroism of (6,5) SWCNT enantiomers [88].

dichroism of (6,5) SWCNT enantiomer suspensions [88] (figure 9*e*) have revealed these excitons, but any quantitative analysis, such as oscillator strength estimation, has not been possible. The crystalline SWCNT films produced by the controlled vacuum filtration technique are compatible with various dopants and electrolyte gating techniques and allow for a direct measurement using macroscopic spectroscopy techniques. These samples give us unambiguous determination of these optical features occurring when light is polarized perpendicular to the nanotube axis.

First, we discuss recently reported direct evidence for ISBPs in gated and aligned SWCNTs through polarization-dependent absorption spectroscopy by Yanagi *et al.*, which elucidated the origin of the observed unknown feature in these previous studies on doped SWCNTs [89]. A film of macroscopically aligned and packed SWCNTs with an average diameter of 1.4 nm, containing both metallic and semiconducting SWCNTs, was prepared using controlled vacuum filtration. The $E_F$ was tuned using electrolyte gating techniques [94,95]. A schematic illustration of the experimental set-up is shown in figure 10*a*. Figure 10*b* shows parallel-polarization optical absorption spectra for the film at three gate voltages, $V_G = -2.0$, 0.0 and $+4.3$ V. At $V_G = 0.0$ V, the interband transitions $S_{11}$ and $S_{22}$ in semiconducting nanotubes and the interband transition $M_{11}$ in metallic nanotubes are clearly observed. As the gate voltage is increased in the positive (negative) direction, electrons (holes) are injected into the nanotubes through formation of an electric double layer on their surfaces. As a result, the $S_{11}$, $S_{22}$ and $M_{11}$ peaks disappear at $V_G = -2.0$ and $+4.3$ V because of Pauli blocking.

Completely different behaviour is observed in perpendicular-polarization spectra, as shown in figure 10*c*. As $V_G$ is increased, a new absorption peak due to ISBP appears at $\sim$1 eV, grows in intensity and dominates the spectrum at the highest $V_G$. The properties of the ISBP peak are entirely orthogonal to those of the $S_{11}$, $S_{22}$ and $M_{11}$ peaks. Figure 10*d* shows detailed changes of the ISBP peak for this aligned film as the gate voltage is varied on the electron injection side. The peak first appears at 1 eV when the gate voltage reaches 3 V, and then its intensity gradually increases with increasing $V_G$. In addition, the peak position blue shifts with further increasing $V_G$, reaching 1.05 eV at 4.3 V.

Second, a systematic polarization-dependent optical absorption spectroscopy study of a macroscopic film of highly aligned single-chirality (6,5) SWCNTs was performed for the observation of crosspolarized excitons [96]. Recent advances in pH-controlled gel chromatography [97,98] enable the preparation of a

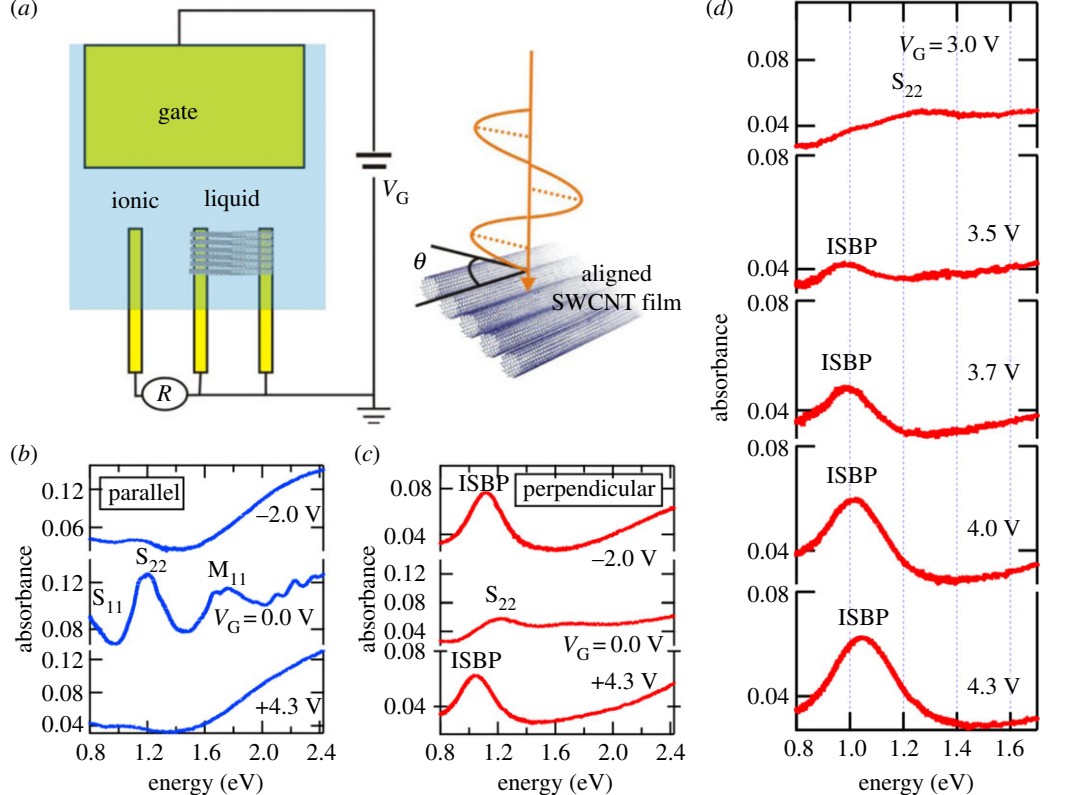

**Figure 10.** ISBPs in gated aligned SWCNT films. (*a*) Schematic diagram for the set-up for the polarization-dependent absorption spectroscopy experiments on an aligned SWCNT film gated through electrolyte gating. $V_G$: gate voltage, $R$: reference voltage. (*b*) Parallel-polarization and (*c*) perpendicular-polarization optical absorption spectra at different gate voltages for an aligned SWCNT film containing semiconducting and metallic nanotubes with an average diameter of 1.4 nm. The ISBP peak appears only for perpendicular polarization for both electron and hole doping, corresponding to $V_G = -2.0$ and 4.3 V, respectively. (*d*) Optical absorption spectra showing how the ISBP peak evolves as the gate voltage increases. Adapted from [89].

large-scale suspension of chirality-enriched semiconducting SWCNTs with extremely high purity. Figure 11*a* shows an example of approximately 30 ml suspension of (6,5) SWCNTs. This industry-scale separation [97] is highly desirable for the vacuum filtration technique to produce aligned films. Figure 11*b* demonstrates an absorption spectrum for the obtained suspension, where only transitions associated with (6,5) SWCNTs are observed, such as $E_{11}$, $E_{22}$, $E_{33}$, and their phonon sidebands. The chirality purity is estimated to be greater than 99% [96,98]. As the polarization of the incident light rotates, the attenuation of the transmitted light changes accordingly. Figure 11*c* displays representative spectra at angles of 0°, 30°, 60° and 90°. Absorption peaks associated with allowed transitions for parallel polarization ($A_\parallel$) are strongly anisotropic. Although there is still prominent absorption at 90° due to imperfect alignment, there is a clearly observable new feature at approximately 1.9 eV in the perpendicular polarization case ($A_\perp$) from the crosspolarized excitons $E_{12}/E_{21}$. Figure 11*d* shows the $E_{12}/E_{21}$ feature more clearly. The baseline was subtracted, and then a spectrum, showing $A_\perp$ multiplied by 3.2 and subtracted by $A_\parallel$, is presented. Further detailed analysis based on nematic order parameters revealed that the oscillator strength for $E_{12}$ ($f_{12}$) is 1/10 of the oscillator strength of $E_{11}$ ($f_{11}$) [96].

# 4. Microcavity exciton-polaritons in aligned semiconducting single-wall carbon nanotubes

Strong coupling of photons and excitons inside a microcavity produces hybrid quasiparticles, microcavity exciton-polaritons, which continue to stimulate much interest [99–101]. In a general model, a two-level system interacts with photons inside a microcavity, with the coupling strength $g$ and photon (matter) decay rate $\kappa$ ($\gamma$). Resonant coupling leads to a splitting into two normal modes with an energy separation, $\hbar\Omega_R$, known as the vacuum Rabi splitting (VRS), with $2g \approx \Omega_R$ when $g \gg \kappa, \gamma$. Depending

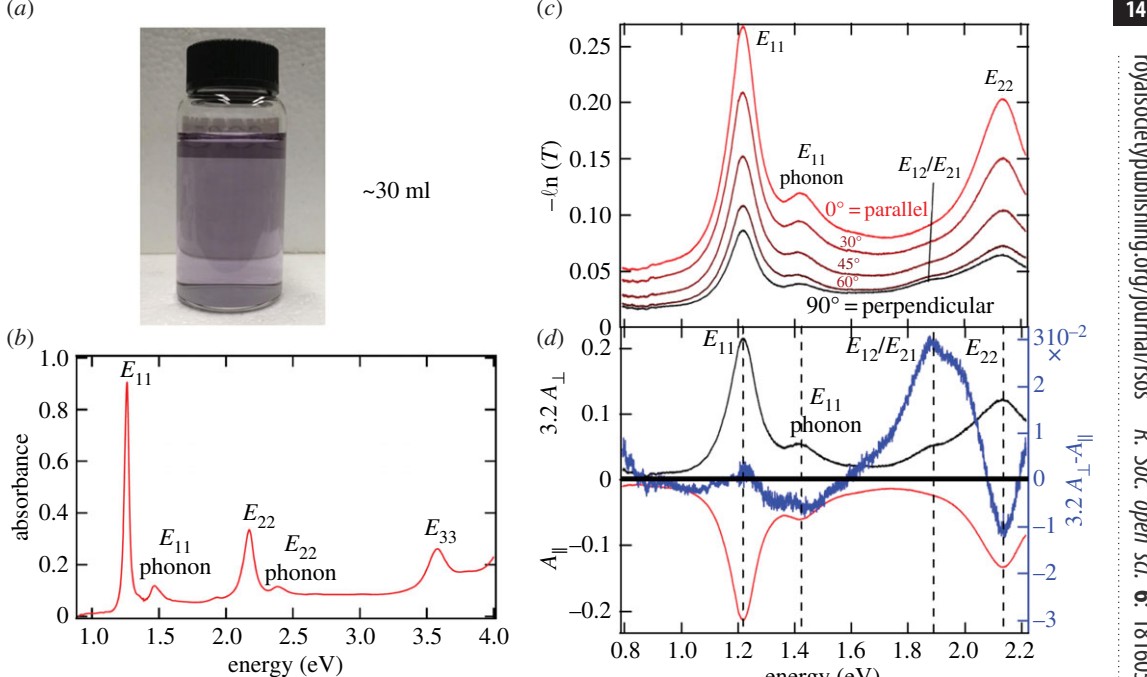

**Figure 11.** Cross-polarized excitons in aligned single-chirality (6,5) SWCNT films. (*a*) A photograph of separated high-purity (6,5) suspension using the gel chromatography technique. (*b*) Absorption spectrum in the near-infrared and visible range for this suspension. The purity is estimated to be greater than 97%. (*c*) Polarization-dependent attenuation spectra for the aligned (6,5) SWCNT film at polarization angles of 0°, 30°, 45°, 60° and 90° with respect to the nanotube alignment direction. (*d*) Comparison of spectra at 0° ($A_\parallel$) and 90° ($A_\perp$). The blue line indicates $3.2A_\perp - A_\parallel$. Adapted from [96].

on the relationship between $g$ and $\kappa$, $\gamma$, there are different regimes [102]. The strong coupling regime is achieved when $4g^2/(\kappa\gamma) > 1$. Furthermore, the ultrastrong coupling regime arises when $g/\omega_0 > 0.1$, where $\omega_0$ is the resonance energy, whereas the deep strong coupling regime is defined as $g/\omega_0 > 1$ [103].

Microcavity exciton-polaritons based on semiconductor quantum wells have been a model system for highlighting the difference between light-atom-coupling and light-condensed-matter-coupling. However, they remain in the strong coupling regime typically with $g/\omega_0 < 10^{-2}$. Furthermore, the fabrication of such devices usually requires sophisticated molecular beam epitaxy, and measurements have to be done at cryogenic temperatures. Excitons in organic semiconductors, possessing large binding energies and oscillator strengths, have displayed larger VRS [104,105] at room temperature. Moreover, nanomaterials, such as transition metal dichalcogenides [106] and semiconducting SWCNTs [107,108], have recently emerged as a new platform for studying strong-coupling physics under extreme quantum confinement. Large oscillator strengths in these materials help relax the stringent requirement for a high quality factor (high-Q) photonic cavity. In particular for SWCNTs, Graf *et al.* first reported near-infrared microcavity exciton-polaritons by incorporating a film of polymer-selected chirality-enriched (6,5) SWCNTs of random orientation inside a Fabry-Pérot microcavity, simply consisting of two parallel metal mirrors [107]. Cavities of similar structures are widely used in organic molecule exciton-polaritons. A very large VRS, exceeding 100 meV, was observed to scale proportionally to $\sqrt{N}$, where $N$ is the number of SWCNTs inside the cavity, displaying a cooperative enhancement [107]. Furthermore, electrical pumping of exciton-polaritons was achieved by the same group [108], paving the way toward carbon-based polariton emitters and lasers.

More recently, Gao *et al.* developed a unique architecture in which excitons in an aligned single-chirality (6,5) CNT film interact with cavity photons in a polarization-dependent manner [109]. Again, they prepared aligned (6,5) films by vacuum filtrating suspensions separated using the aqueous two-phase extraction method. The microcavity exciton-polariton devices were created by embedding the obtained aligned film with thickness $d$ inside a Fabry–Pérot cavity. By changing the incidence angle ($\theta$ in figure 12a), the resonance frequency of the microcavity was tuned to resonate with the $E_{11}$ or $E_{22}$ of (6,5) SWCNTs. Furthermore, adjustment of the angle between the incident light polarization direction and the SWCNT alignment direction ($\phi$ in figure 12a) provided a convenient knob to probe the directional variation of $g$. The angle $\theta$ has a one-to-one correspondence with the in-plane wavevector $k_\parallel$, so the $\theta$ dependence of

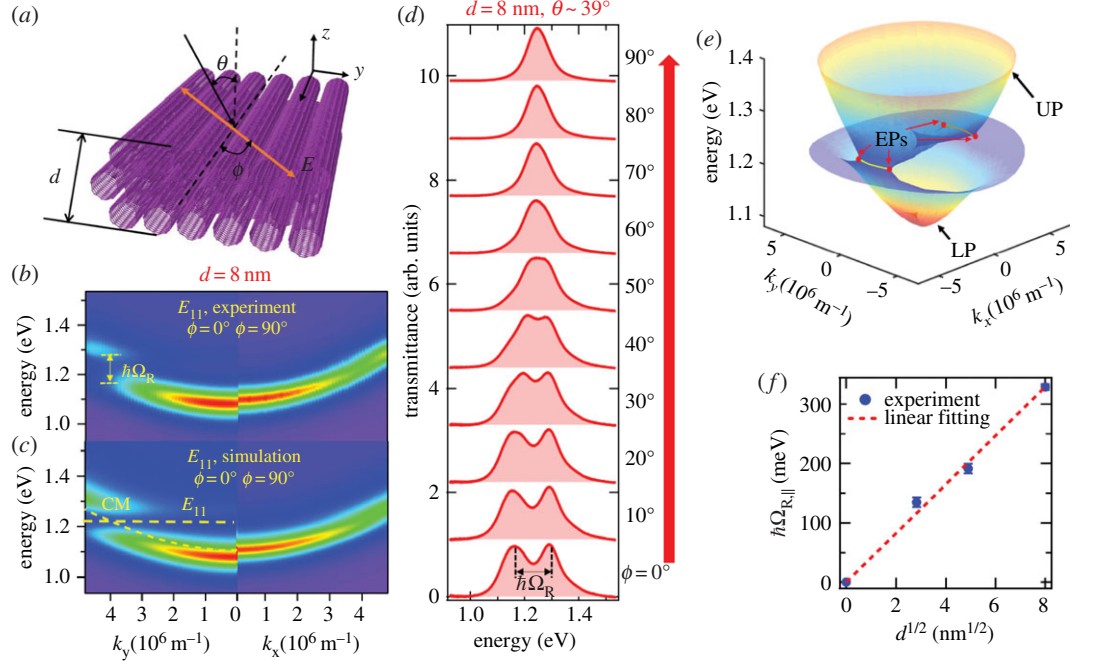

**Figure 12.** Microcavity exciton polaritons based on aligned semiconducting SWCNTs. (a) The two angles that are continuously scanned in the present experiments—the incident angle $\theta$ and the polarization angle $\phi$. The latter is the angle between the SWCNT alignment direction (x-axis) and the incident light polarization direction. (b) Anisotropic dispersions of microcavity exciton-polaritons in the $E_{11}$ region for $\phi = 0°$ and $\phi = 90°$. A clear VRS is observed at $\phi = 0°$, while a photon dispersion with no splitting is observed at $\phi = 90°$. (c) Corresponding simulated anisotropic dispersions in the $E_{11}$ region. (d) Experimental transmittance spectra at zero detuning ($\theta \sim 39°$) for various polarization angles $\phi$ from 0° to 90° for a device working in the $E_{11}$ region using an aligned (6,5) SWCNT film with $d = 8$ nm. (e) Dispersion surfaces of the upper polariton (UP) and lower polariton (LP) (coloured surfaces) for the device in (d). (f) VRS at $\phi = 0°$ versus the square root of the film thickness, demonstrating the $\sqrt{N}$-fold enhancement of collective light-matter coupling. Adapted from [109].

transmission peaks can be converted into $k_y$ or $k_x$ dependence for $\phi = 0°$ or 90°, respectively, to map out polariton dispersions. Figure 12b,c shows experimental and simulated in-plane dispersion relationships for a device with $d = 8$ nm for $\phi = 0°$ and 90°, respectively. A prominent VRS of $137 \pm 6$ meV ($g/\omega_0 > 5.5\%$) is observed, while no splitting is seen at $\phi = 90°$. Here, $\omega_0$ is the resonance frequency of $E_{11}$. Furthermore, any value of VRS between zero and its maximum value can be selected, on demand, by setting the polarization angle $\phi$ (figure 12d).

As the in-plane wavevector ($k_x$, $k_y$) has one-to-one correspondence with ($\phi$, $\theta$), the continuous adjustment of $\phi$ and $\theta$ maps out the full anisotropic SWCNT exciton polariton dispersion surfaces, as demonstrated in figure 12e for the 8-nm-thick device. The most striking feature of figure 12e is the appearance of two circular arcs on which the energy is constant; namely, the upper polariton (UP) and lower polariton (LP) branches coalesce to form these constant-energy arcs in momentum space. Furthermore, Gao et al. analysed the obtained spectra using quantum Langevin equations, including losses, and demonstrated that the end points of each arc are exceptional points, that is, spectral singularities that ubiquitously appear in open or dissipative systems. Finally, figure 12f confirms that VRS scales with $\sqrt{d}$ and thus $\sqrt{N}$. The VRS value for the 64-nm-thick device was 329 meV ($g/\omega_0 > 13.3\%$), which is the largest value ever reported for exciton polaritons based on Wannier excitons.

# 5. Plasmon resonances and hybridization in patterned aligned single-wall carbon nanotubes

The longitudinal confinement of free carriers in finite-length SWCNTs leads to a prominent broad absorption peak in the THz and infrared ranges [66,72], as shown in figure 6b. Temperature-dependent broadband spectroscopy unambiguously revealed the plasmonic nature in films of randomly oriented SWCNTs [72]. The resonance energy depends on the length of the SWCNT and thus can be tuned by adjusting the nanotube length through sonication duration [110]. However, due to a broad length

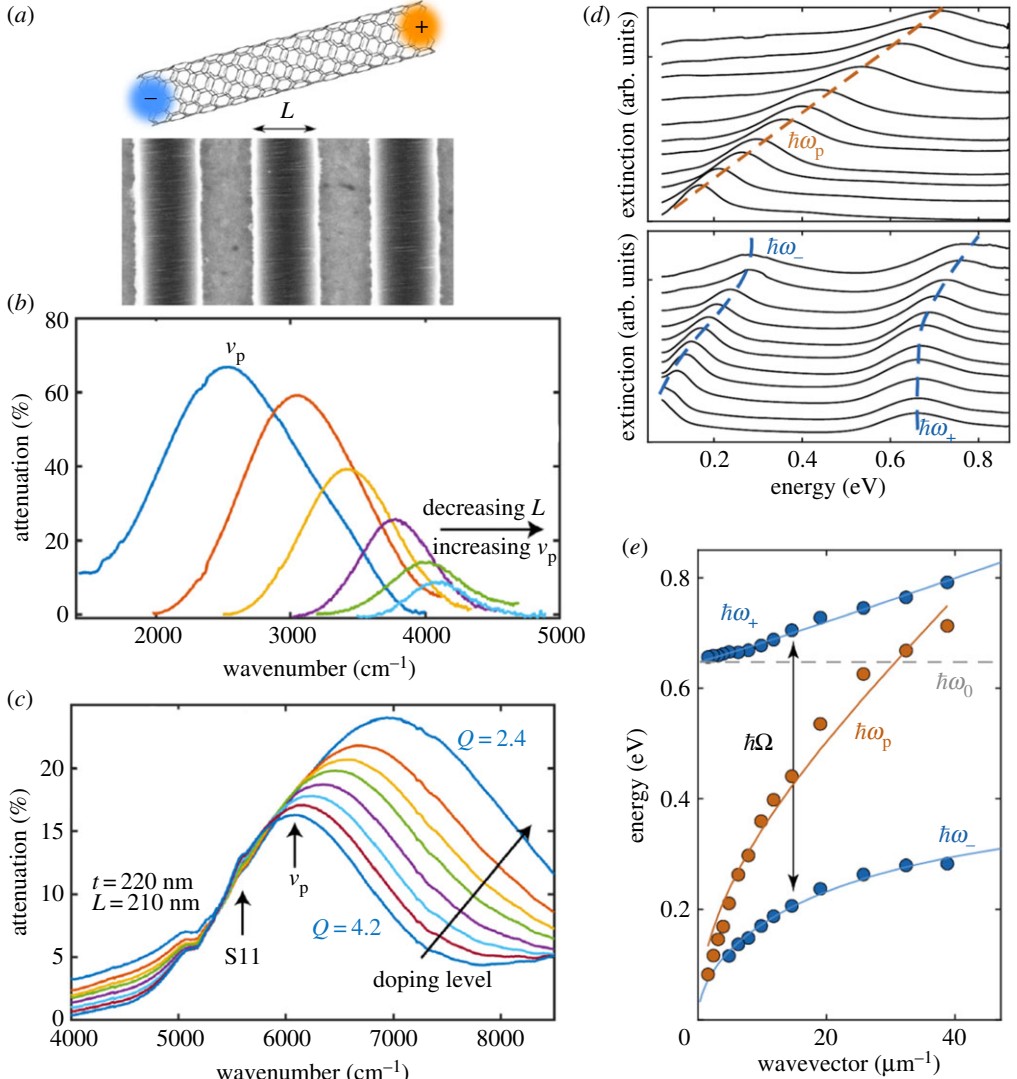

**Figure 13.** Plasmon resonances and plexcitons in patterned aligned SWCNTs. (*a*) Schematic diagram of plasmon resonances in patterned aligned SWCNTs. An SEM image shows a fabricated device. (*b*) Attenuation spectra for strips of aligned SWCNTs with different *L*. The peak energy is $v_p$. (*c*) Attenuation spectra for strips of aligned SWCNTs with different doping levels. (*d*) The extinction spectra of patterned films of highly doped aligned SWCNTs (top) and annealed aligned SWCNTs (bottom). (*e*) Resonance energies as a function of wavevector $q$, which is defined as $\pi/L$. The solid line is a fit to $\omega_p \propto \sqrt{q}$ (orange lines). Peak energies of the plexcitons, showing an anticrossing (blue lines). Adapted from Ref. [71] and Ref. [69].

distribution in these films, the linewidth of the resonance is usually broad, and the resonance peak position is up to approximately $400\ \mathrm{cm}^{-1}$ in the far-infrared. Crystalline SWCNT films produced using the controlled vacuum filtration method, which are compatible with various micro/nanofabrication techniques such as lithography and etching, provide a unique platform for unifying CNT lengths for high-quality plasmon resonators. The longitudinal confinement, determined by the lithography-defined dimensions, can range from tens to hundreds of nanometres. In combination with chemical doping techniques, the resonance frequency can be broadly tuned from the THz to near-infrared.

Figure 13*a* illustrates the bounded electron oscillation in SWCNTs and a representative SEM image of patterned strips of aligned films of SWCNTs produced through controlled vacuum filtration reported by Chiu *et al.* [71]. For strips of a 165-nm-thick film, there are strong attenuation peaks ($v_p$) in the midinfrared range. As the lateral dimension (*L*) decreases from 800 nm to 100 nm at fixed film thickness (*t*), $v_p$ blue-shifts and *Q* increases up to 9 (figure 13*b*). This increase of the *Q* factor reflects the increased length uniformity in smaller-*L* devices. Furthermore, exposure to strong dopants, such as HNO₃, increases the film conductivity and pushes $v_p$ further to the near-infrared, as shown in figure 13*c*. The presence of dopants, however, reduces the *Q* factor from 4.2 to 2.4, because of an increase of scattering.

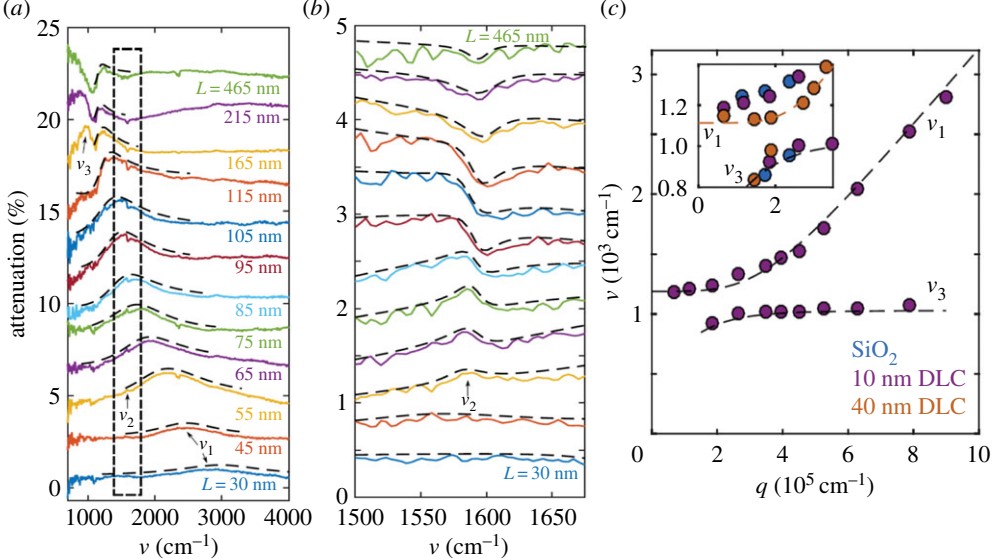

**Figure 14.** Phonon-plasmon-polaritons in patterned aligned SWCNTs. (*a*) The attenuation spectra of strips of aligned SWCNTs with different lengths on the 40-nm diamondlike carbon substrate. (*b*) An expanded view of the dashed rectangle in figure 14*a*, along with fits to Fano functions (black dashed lines). (*c*) The resonant frequencies, $\nu_1$ and $\nu_3$, as a function of $q$, on substrates with different thicknesses of diamondlike carbon. Adapted from [111].

As dopants are introduced and the $E_F$ is elevated, the lowest-energy excitons in semiconducting SWCNTs, $S_{11}$, in these films are switched off (figures 8*g* and 9*a*). There is a single resonance peak as $L$ is tuned, as shown in the top panel of figure 13*e*. After films are annealed to remove excess dopants, exciton transitions are activated with the $S_{11}$ transition energy at $\omega_0 = 0.66\,\text{eV}$. As $L$ is continuously adjusted and thus $\nu_p$ is continuously tuned across $\omega_0$, strong coupling between plasmon and exciton resonances emerges; see the bottom panel of figure 13*d*. The formed new quasi-particles are known as plexcitons. Figure 13*e* demonstrates the peak energies for the LP branch, UP branch and bare plasmon resonances in highly doped films. An extremely large VRS $\hbar\Omega \sim 485\,\text{meV}$ ($g/\omega_0 \sim 36.7\%$) is observed, indicating ultrastrong coupling. A Hopfield-like Hamiltonian can fully capture the polariton dispersions [69]. Moreover, the coupling strength can be tuned by adjusting the doping level and film thickness. The increase of doping level, thus film conductivity, decreases the oscillator strength of $S_{11}$ and the coupling strength.

In addition to plexcitons, these plasmon resonances in strips of aligned SWCNTs can strongly couple with other quasi-particles, such as phonons. Falk *et al.* showed strong coupling between longitudinal optical phonons in SiO$_2$ substrates (attenuation peak $\nu_1$ in figure 14*a*), infrared-active $E_1$ and $E_{1u}$ phonons in CNTs [112,113] (attenuation peak $\nu_2$ in figure 14*a*) and plasmon resonances (attenuation peak $\nu_3$ in figure 14*a*), by putting patterned strips with various $L$ of aligned SWCNT films on SiO$_2$ substrates. The strong coupling enhances usually weak infrared-active phonons and leads to an asymmetric lineshape, which can be fit very well with a Fano resonance formula [111], as shown in figure 14*b*. Peak positions of $\nu_1$ and $\nu_3$ demonstrate a clear anticrossing behaviour, as displayed in figure 14*c*.

# 6. Conclusion and challenges

This review summarizes the recent successes in fabricating wafer-scale crystalline CNT films and the new science and applications that these films have enabled, specifically in photonics and optoelectronics. These wafer-scale crystalline CNTs with a controlled chirality distribution fabricated through controlled vacuum filtration open new pathways toward both fundamental studies and applications in diverse disciplines. This technique addresses the grand 'bottom-to-top' challenge in the CNT community for utilizing the extraordinary properties of individual CNTs at the nanoscale in macroscopic applications. We anticipate that the initial studies described in this review will further stimulate interest in CNT research and push toward real-world applications.

The future direction of improving the controlled vacuum filtration technique can focus on addressing challenges in the current process. To have an optimal alignment quality, extensive tip sonication has to be employed to reduce the CNT length and increase the stiffness, because kinks, joints and defects in long tubes make them vulnerable to bending and twisting in dispersions [114,115]. These short tubes in assembled aligned films introduce a number of tube–tube junctions, which dominate and limit the transport performance of electronic devices. One direction is to explore different dispersion solvents for preserving long tubes with increased stiffness, and search for different surface chemistry of filter membranes and optimal filtration conditions due to altered colloidal properties of dispersed CNTs. Furthermore, although the thickness of obtained films prepared using controlled vacuum filtration can range from a few nanometres to a few microns, it is challenging to prepare monolayer films of aligned CNTs with adjustable spacing, which are necessary for high-performance electronics applications. One possible way of solving this issue is to reduce the thickness through layer-by-layer etching.

Data accessibility. This article has no additional data.

Authors' contributions. W. G. and J. K. wrote the manuscript and gave the final approval for publication.

Competing interests. We have no competing interests.

Funding. This work was supported by the Department of Energy Basic Energy Sciencesthrough grant no. DE-FG02-06ER46308, the National Science Foundationthrough Award No. ECCS-1708315, and the Robert A. Welch Foundationthrough grant no. C-1509.

Acknowledgements. We thank G. Timothy Noe for his help with editing and proofreading a draft version of this article. Some of the work presented in this article was originally carried out in collaboration with Xiaowei He, Fumiya Katsutani, Xinwei Li and Natsumi Komatsu from Rice University, and Kazuhiro Yanagi from Tokyo Metropolitan University.

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
