## [Reviewer comments · Royal Society Open Science]

Review History

RSOS-181605.R0 (Original submission)

Review form: Reviewer 1

Is the manuscript scientifically sound in its present form?

Yes

Are the interpretations and conclusions justified by the results?

Yes

Is the language acceptable?

Yes

Is it clear how to access all supporting data?

Not Applicable

Do you have any ethical concerns with this paper?

No

Have you any concerns about statistical analyses in this paper?

No

Recommendation?

Accept with minor revision (please list in comments)

Comments to the Author(s)

The authors reviewed the Science and applications of wafer-scale crystalline carbon nanotube films prepared through controlled vacuum filtration. This review summarize recent discoveries in optical spectroscopy studies and optoelectronic device applications using films prepared by this technique. This work is systematic and comprehensive, and so I suggest this manuscript can be published after completing the following revisions:

1. The authors should further discuss the challenge of wafer-scale crystalline carbon nanotube films prepared through controlled vacuum filtration.

2. The relevant literatures about carbon nanotube films are suggested to be referred to, such as Sensors and Actuators B: Chemical, v 258, p 895-905, 2018; Journal of Materials Science: Materials in Electronics, v 26, n 10, p 7445-7451, 2015; Microsystem Technologies, v 20, n 3, p 379-385, 2014; Microsystem Technologies, v 19, n 7, p 1041-1047, 2013; Sensors and Actuators A: Physical, v 185, p 101-108, 2012.

Review form: Reviewer 2 (Lian-Mao Peng)

Is the manuscript scientifically sound in its present form?

Yes

Are the interpretations and conclusions justified by the results?

Yes

Is the language acceptable?

Yes

Is it clear how to access all supporting data?

Not Applicable

Do you have any ethical concerns with this paper?

No

Have you any concerns about statistical analyses in this paper?

No

Recommendation?

Accept with minor revision (please list in comments)

Comments to the Author(s)

This is an excellent review on the preparation of wafer-scale CNT films and some examples of applications of these films. The manuscript is already well prepared. But for the completeness, I would suggest the following few new papers for the consideration of the authors. (1) Jiantao

Wang et al., Nature Catalysis, (2018) DOI:10.1038/s41929-018-0057-x, Growing highly pure semiconducting carbon nanotubes by electrotwisting the helicity; (2) Jia Si et al., ACS Nano 2018, 12:627-634, Scalable preparation of high density semiconducting carbon nanotube arrays for high performance field-effect transistors; (3) M. Zhu et al., Adv. Mater. 2018, 30:1707068, Aligning solution-derived carbon nanotube film with full surface coverage for high performance electronics applications. These methods are somehow different from usual chemical methods, and may potentially be used for digital electronics.

While it has been demonstrated in the manuscript several photonics and optoelectronics applications, I would be more interested to see some discussions on potential application of the CNT films in digital electronics. It is true that films with different thickness can be obtained, but what is the low limit of the thickness. The most interesting case would be a monolayer CNTs, with somehow large spacing between them than those demonstrated in the paper. Can we control the tube-tube junction in this case, from e.g. 5-10 nm? I know it is difficult, and I do not insist the authors to answer these questions.

Decision letter (RSOS-181605.R0)

03-Jan-2019

Dear Dr Gao:

Title: Science and applications of wafer-scale crystalline carbon nanotube films prepared through controlled vacuum filtration
Manuscript ID: RSOS-181605

Thank you for submitting the above manuscript to Royal Society Open Science. On behalf of the Editors and the Royal Society of Chemistry, I am pleased to inform you that your manuscript will be accepted for publication in Royal Society Open Science subject to minor revision in accordance with the referee suggestions. Please find the reviewers' comments at the end of this email. I apologise for the delay.

The reviewers and handling editors have recommended publication, but also suggest some minor revisions to your manuscript. Therefore, I invite you to respond to the comments and revise your manuscript.

Please also include the following statements alongside the other end statements. As we cannot publish your manuscript without these end statements included, if you feel that a given heading is not relevant to your paper, please nevertheless include the heading and explicitly state that it is not relevant to your work. We have included a screenshot example of the end statements for reference.

- Ethics statement

Please clarify whether you received ethical approval from a local ethics committee to carry out your study. If so please include details of this, including the name of the committee that gave consent in a Research Ethics section after your main text. Please also clarify whether you received informed consent for the participants to participate in the study and state this in your Research Ethics section.

OR

Please clarify whether you obtained the necessary licences and approvals from your institutional animal ethics committee before conducting your research. Please provide details of these licences and approvals in an Animal Ethics section after your main text.

OR

Please clarify whether you obtained the appropriate permissions and licences to conduct the fieldwork detailed in your study. Please provide details of these in your methods section.

- Authors' contributions

Please include an Authors' Contributions section at the end of your main text detailing the contribution of each author. All authors should have read and approved the manuscript before submission and this should be stated in the Authors' Contributions section.

The list of Authors should meet all of the following criteria; 1) substantial contributions to conception and design, or acquisition of data, or analysis and interpretation of data; 2) drafting the article or revising it critically for important intellectual content; and 3) final approval of the version to be published.

Because the schedule for publication is very tight, it is a condition of publication that you submit the revised version of your manuscript before 12-Jan-2019. Please note that the revision deadline will expire at 00.00am on this date. If you do not think you will be able to meet this date please let me know immediately.

- 1) A text file of the manuscript (tex, txt, rtf, docx or doc), references, tables (including captions) and figure captions. Do not upload a PDF as your "Main Document".
- 2) A separate electronic file of each figure (EPS or print-quality PDF preferred (either format should be produced directly from original creation package), or original software format)
- 3) Included a 100 word media summary of your paper when requested at submission. Please ensure you have entered correct contact details (email, institution and telephone) in your user account

4) Included the raw data to support the claims made in your paper. You can either include your data as electronic supplementary material or upload to a repository and include the relevant doi within your manuscript

5) All supplementary materials accompanying an accepted article will be treated as in their final form. Note that the Royal Society will neither edit nor typeset supplementary material and it will be hosted as provided. Please ensure that the supplementary material includes the paper details where possible (authors, article title, journal name).

Best wishes,
Dr Laura Smith
Publishing Editor, Journals

On behalf of the Subject Editor Professor Anthony Stace and the Associate Editor Professor Claire Carmalt.

RSC Associate Editor:
Comments to the Author:
(There are no comments.)

RSC Subject Editor:
Comments to the Author:
(There are no comments.)

Reviewer comments to Author:
Reviewer: 1

Comments to the Author(s)
The authors reviewed the Science and applications of wafer-scale crystalline carbon nanotube films prepared through controlled vacuum filtration. This review summarize recent discoveries in optical spectroscopy studies and optoelectronic device applications using films prepared by

this technique. This work is systematic and comprehensive, and so I suggest this manuscript can be published after completing the following revisions:

1. The authors should further discuss the challenge of wafer-scale crystalline carbon nanotube films prepared through controlled vacuum filtration.
2. The relevant literatures about carbon nanotube films are suggested to be referred to, such as *Sensors and Actuators B: Chemical*, v 258, p 895-905, 2018; *Journal of Materials Science: Materials in Electronics*, v 26, n 10, p 7445-7451, 2015; *Microsystem Technologies*, v 20, n 3, p 379-385, 2014; *Microsystem Technologies*, v 19, n 7, p 1041-1047, 2013; *Sensors and Actuators A: Physical*, v 185, p 101-108, 2012.

Reviewer: 2

Comments to the Author(s)

This is an excellent review on the preparation of wafer-scale CNT films and some examples of applications of these films. The manuscript is already well prepared. But for the completeness, I would suggest the following few new papers for the consideration of the authors. (1) Jiantao Wang et al., *Nature Catalysis*, (2018) DOI:10.1038/s41929-018-0057-x, Growing highly pure semiconducting carbon nanotubes by electrotwisting the helicity; (2) Jia Si et al., *ACS Nano* 2018, 12:627-634, Scalable preparation of high density semiconducting carbon nanotube arrays for high performance field-effect transistors; (3) M. Zhu et al., *Adv. Mater.* 2018, 30:1707068, Aligning solution-derived carbon nanotube film with full surface coverage for high performance electronics applications. These methods are somehow different from usual chemical methods, and may potentially be used for digital electronics.

While it has been demonstrated in the manuscript several photonics and optoelectronics applications, I would be more interested to see some discussions on potential application of the CNT films in digital electronics. It is true that films with different thickness can be obtained, but what is the low limit of the thickness. The most interesting case would be a monolayer CNTs, with somehow large spacing between them than those demonstrated in the paper. Can we control the tube-tube junction in this case, from e.g. 5-10 nm? I know it is difficult, and I do not insist the authors to answer these questions.

Author's Response to Decision Letter for (RSOS-181605.R0)

See Appendix A.

Decision letter (RSOS-181605.R1)

06-Feb-2019

Dear Dr Gao:

Title: Science and applications of wafer-scale crystalline carbon nanotube films prepared through controlled vacuum filtration

Manuscript ID: RSOS-181605.R1

It is a pleasure to accept your manuscript in its current form for publication in Royal Society Open Science. The chemistry content of Royal Society Open Science is published in collaboration with the Royal Society of Chemistry.

On behalf of the Subject Editor Professor Anthony Stace and the Associate Editor Professor Claire Carmalt.

RSC Associate Editor
Comments to the Author:
(There are no comments.)

Reviewer(s)' Comments to Author:

Appendix A

Response to Reviewer #1

Reviewer #1's comment #1: The authors should further discuss the challenge of wafer-scale crystalline carbon nanotube films prepared through controlled vacuum filtration.

Response to comment #1: We agree with the reviewer that we should mention the challenges we have for producing wafer-scale crystalline carbon nanotube films. Specifically, one of the challenges is that our method requires extensive tip sonication to reduce the carbon nanotube length and increase the stiffness for optimal alignment quality, because kinks, joints, and defects in long tubes make them vulnerable to bending and twisting in the dispersion. Also, it still remains a challenge to use the controlled vacuum filtration technique to produce a monolayer film of aligned carbon nanotubes with a controllable tube spacing, required for high-performance electronics. **In the conclusion section (now conclusion and challenges) of the revised manuscript, we have added a new paragraph discussing these two challenges.**

Reviewer #1's comment #2: The relevant literatures about carbon nanotube films are suggested to be referred to, such as *Sensors and Actuators B: Chemical*, v 258, p 895-905, 2018; *Journal of Materials Science: Materials in Electronics*, v 26, n 10, p 7445-7451, 2015; *Microsystem Technologies*, v 20, n 3, p 379-385, 2014; *Microsystem Technologies*, v 19, n 7, p 1041-1047, 2013; *Sensors and Actuators A: Physical*, v 185, p 101-108, 2012.

Response to comment #2: The papers mentioned by the reviewer focus on composite films, which include carbon nanotubes and other materials, and their potential applications in nano/micro-electro-mechanical systems and gas/liquid sensors. Although these composite films and applications digress from the main topic of this review, we agree with the referee that it would be worthwhile mentioning some of these papers in the introduction. **At the end of the 5th paragraph of subsection (a) in the introduction part of the revised manuscript, we have added a sentence mentioning these two potential applications and two selected papers (*Journal of Materials Science: Materials in Electronics*, v 26, n 10, p 7445-7451, 2015 and *Sensors and Actuators A: Physical*, v 185, p 101-108, 2012) in the reference.**

Response to Reviewer #2

Reviewer #2's comment #1: But for the completeness, I would suggest the following few new papers for the consideration of the authors. (1) Jiantao Wang et al., Nature Catalysis, (2018) DOI:10.1038/s41929-018-0057-x, Growing highly pure semiconducting carbon nanotubes by electrotwisting the helicity; (2) Jia Si et al., ACS Nano 2018, 12:627-634, Scalable preparation of high density semiconducting carbon nanotube arrays for high performance field-effect transistors; (3) M. Zhu et al., Adv. Mater. 2018, 30:1707068, Aligning solution-derived carbon nanotube film with full surface coverage for high performance electronics applications. These methods are somehow different from usual chemical methods, and may potentially be used for digital electronics.

Response to comment #1: We thank the reviewer for bringing our attention to these recent papers. We agree that these papers represent important recent progress in preparing aligned semiconducting (but not single-chirality) carbon nanotubes, especially for high-performance electronics applications. The first paper (Nature Catalysis, (2018) DOI:10.1038/s41929-018-0057-x) describes a method of controlling the polarity of catalyst particle charge using an external electric field to twist the chirality of carbon nanotubes from metallic to semiconducting during a chemical vapor deposition process. The second (ACS Nano 2018, 12:627-634) and third (Adv. Mater. 2018, 30:1707068) papers present a straightforward method of directional shrinking transfer to increase the packing density in aligned carbon nanotubes and decrease the angle deviation in randomly oriented carbon nanotubes for creating partially aligned carbon nanotube films. **In the revised manuscript, we have added all three papers in the reference. In addition, we have added one sentence describing the first paper in the 3rd paragraph of subsection (b) in the introduction part of the revised manuscript. We have also added several sentences in the 5th paragraph and one subfigure in Figure (4) to discuss the second and third papers.**

Reviewer #2's comment #2: While it has been demonstrated in the manuscript several photonics and optoelectronics applications, I would be more interested to see some discussions on potential application of the CNT films in digital electronics. It is true that films with different thickness can be obtained, but what is the low limit of the thickness. The most interesting case would be a monolayer CNTs, with somehow large spacing between them than those

demonstrated in the paper. Can we control the tube-tube junction in this case, from e.g. 5-10 nm? I know it is difficult, and I do not insist the authors to answer these questions.

Response to comment #2: We thank the reviewer for this valuable comment. The achieved lowest limit of the film thickness is ~5 nm, and it still cannot be treated as a monolayer film of carbon nanotubes with typical diameter ~1 nm. We agree with the reviewer that monolayer films with medium spacing (not dense packing) would be ideal for high-performance electronics, but it is challenging to obtain such films using our controlled vacuum filtration technique. Instead, controlled vacuum filtration is more suitable for producing reasonably thick and densely packed films for photonic and optoelectronic applications, which is the main topic of this review. In addition, in order to have an optimal alignment quality, extensive tip sonication has to be employed to reduce the CNT length and increase the stiffness, because kinks, joints, and defects on long tubes make them vulnerable to bending and twisting in dispersion. These short tubes in assembled aligned films introduce a number of tube-tube junctions, which dominate and limit the transport performance of electronic devices. Thus, potential electronic applications based on films prepared by controlled vacuum filtration could be limited. **In the conclusion section (now conclusion and challenges) of the revised manuscript, we have added a new paragraph further discussing the challenges of our vacuum filtration technique and providing possible ways of addressing these challenges, especially from the perspective of electronic applications.**

Changes Made

4th page, 1st paragraph, last sentence

Added: CNTs can also form composites with other materials, such as organic polymers and inorganic nanorods, to reinforce polymer mechanical properties for nano/micro-electro-mechanical systems [26] and build sensitive gas sensors [27].

5th page, 4th paragraph, 3rd sentence

Previous: Recently, there have been breakthroughs in precisely controlling chirality distribution during synthesis, by designing a catalyst that matches the carbon atom arrangement around the nanotube circumference of a specific chirality [37-39].

Current: Recently, there have been breakthroughs in precisely controlling chirality distribution during synthesis, by designing a catalyst that matches the carbon atom arrangement around the nanotube circumference of a specific chirality [37-39] or applying an external electric field to control the polarity of catalyst charge for changing the chirality from metallic to semiconducting [40].

5th page, 6th paragraph, 2nd sentence

Added: Metallic CNTs in horizontally aligned CNTs prepared using epitaxy CVD growth can be selectively removed, leaving an array of high-purity semiconducting CNTs [52]. However, the packing density is as low as 3 CNTs/ μm . A simple directional shrinkage method can compress films to amplify the packing density (Figure 4(b)) [53]. Although this level of low packing density is desirable for high-performance electronics applications, such densities are still too low for photonic and optoelectronic applications. The same technique can be applied to randomly oriented CNT films to reduce the angle deviation for preparing partially aligned films [54].

Figure 4

Added: Added a new subfigure, Figure 4(b), and modified the caption correspondingly.

19th – 20th page, final paragraph

Added: The future direction of improving the controlled vacuum filtration technique can focus on addressing challenges in the current process. In order to have an optimal alignment quality, extensive tip sonication has to be employed to reduce the CNT length and increase the stiffness, because kinks, joints, and defects in long tubes make them vulnerable to bending and twisting in dispersions [114,115]. These short tubes in assembled aligned films introduce a number of tube-tube junctions, which dominate and limit the transport performance of electronic devices. One direction is to explore different dispersion solvents for preserving long tubes with increased stiffness, and search for different surface chemistry of filter membranes and optimal filtration condition due to altered colloidal properties of dispersed CNTs. Furthermore, although the thickness of obtained films prepared using controlled vacuum filtration can range from a few nanometers to a few microns, it is challenging to prepare monolayer films of aligned CNTs with adjustable spacing, which are ideal for high-performance electronics applications. One possible way of solving this issue is to reduce the thickness through layer-by-layer etching.